# Open Data for Open Questions in Comparative Nutrition

**DOI:** 10.3390/insects11040236

**Published:** 2020-04-09

**Authors:** Juliano Morimoto, Mathieu Lihoreau

**Affiliations:** 1School of Biological Sciences, University of Aberdeen, Zoology Building, Tillydrone Ave, Aberdeen AB24 2TZ, UK; 2Research Center on Animal Cognition (CRCA), Center for Integrative Biology (CBI); CNRS, University Paul Sabatier, 31062 Toulouse, France; mathieu.lihoreau@univ-tlse3.fr

**Keywords:** nutritional ecology, geometric framework for nutrition, performance landscape, open data

## Abstract

Achieving a better understanding of the consequences of nutrition to animal fitness and human health is a major challenge of our century. Nutritional ecology studies increasingly use nutritional landscapes to map the complex interacting effects of nutrient intake on animal performances, in a wide range of species and ecological contexts. Here, we argue that opening access to these hard-to-obtain, yet considerably insightful, data is fundamental to develop a comparative framework for nutrition research and offer new quantitative means to address open questions about the ecology and evolution of nutritional processes.

## 1. The Multidimensional Nature of Nutrition

The World Health Organisation (WHO) estimates that ca. 2.4 billion adults experience health-related problems due to malnutrition (https://www.who.int/en/news-room/fact-sheets/detail/malnutrition). Such alarming numbers have prompted scientists, clinicians, and policy makers to develop a more integrative nutrition science, by focusing their research on the effects of nutrition across a wide variety of contexts, spanning from fundamental science involving animal nutrition and the consequences to animal behaviour and evolution to human disease [1,2].

In recent decades, the conceptual framework known as the geometric framework for nutrition (GFN; Figure 1) [3] has revolutionized research in nutritional ecology by taking into account how variation in the quantity and balance of nutrients in foods can affect the development, metabolic health, reproduction and ageing of animals [2]. The GFN allows for direct experimental assessment of the effects of multiple nutrients and their calorie intakes on performance traits and ultimately fitness expression (e.g., Figure 1A,B) [4]. This has brought key insights into the nutritional factors of a wide variety of physiological and behavioural phenomena across feeding guilds (e.g., herbivores, carnivores, and omnivores) and taxonomic groups (e.g., slime moulds, insects, fish, birds, and mammals) [2,5]. These progresses in nutrition research also have far reaching implications in various areas of biology, ecology and evolution, including to our understanding of contemporary diseases affecting human societies [6]. Here, we argue that making GFN data open access has considerable benefits to advancing the broad field of nutrition research and tackling open questions on the ecology and evolution of nutritional processes through quantitative comparative approaches of complex nutritional systems across ecological contexts and scales of organisations. To do so, we provide an up-to-date summary of available data and a guideline for the standardized sharing of future data.

## 2. The Geometric Framework for Nutrition in a Nutshell

The GFN [2] is a state-space modelling approach that can be used to study the simultaneous contribution of two or more nutrients to performance traits (e.g., ratios of proteins to carbohydrates [7], lipids to proteins [8], proteins to lipids and carbohydrates [9,10], and amino acids [11]). Diets with different nutrient ratios (i.e., ‘nutritional rails’, Figure 1A) determine the ‘width’ of the performance landscape in the nutritional plane (i.e., ‘nutritional space’, Figure 1A). Diets with the same nutrient ratio can have different nutrient concentrations, which introduces variance in individual nutrient intake and determines the ‘length’ of the performance landscape (Figure 1A). Experimentally, the higher the number of different nutrient ratios, nutrient concentrations and replicates, the higher the resolution of the landscape (see [2] for a review).

Higher resolution, and thus confidence in the results, is obtained by a higher coverage of the nutritional space with more nutritional rails which are also in closer proximity, following:
$$\frac{90{}^{\circ}}{N_{rails}}=\emptyset$$
where $N_{rails}$ is the number of nutritional rails in the experimental design and ∅ is the average distance between rails. However, an increase in the number of nutrient ratios in arithmetic scale also results in an increase in sample size in the geometric scale, following:
$$N_{final}=NR\times C\times R$$
where *N_final_* is the final sample size of the experiment, *NR* is the number of nutrient ratios, *C* is the number of nutrient concentrations, and *R* is the number of replicates per concentration per ratio. This relationship constrains experimental designs that can grow rapidly and compromises their feasibility. For example, GFN experiments tend to adopt a design of at least five replicates of five nutrient ratios with three concentrations each (75 replicates) [2]. For an experiment with no more than six nutrient ratios, the sample size increases from 75 to 90 replicates (6 × 3 × 5) for an increase of ~17% in the landscape resolution (15° between nutritional rails). Data collection is therefore a bottleneck for the quality and resolution of fitness landscape and represents a trade-off between feasibility (i.e., the rapid increase in sample size) and resolution (i.e., higher coverage of the nutritional space).

For field studies, when this approach can be tedious and all required information may not be available, a derivative of the GFN known as the Right-Angle Mixture Triangle (RMT) [13] can be used to transform diet contents into relative percentages of each specific nutrient of interest. The percentage of each nutrient in the diet is represented in the axes of RMT (Figure 1C), which depicts the nutrient mixture in percentage of the diet. For example, a recent study analysed the macronutrient composition of mammalian milk, revealing that, overall, primate milk is lower in protein concentration relative to the milk of other mammals (Figure 1D) [12]. Since nutrient mixtures are represented by percentages, the concentration of the nutrients is not incorporated into the graphical representation which instead offers a graphical overview of the diet mixture (see Figure 1A). Nonetheless, in theory, RMT can be used to derive multidimensional performance landscapes and compare broader nutritional ecology patterns between populations in different environments or between species occupying the same environment [13]. Here, data collection is also a bottleneck for data resolution and statistical inferences in comparative studies, meaning that this approach also experiences a trade-off on quality (and costs) and resolution.

For the remainder of this paper, we will use the GFN to refer to both the more general GFN (described above) and the derivative RMT. All concepts proposed here are therefore applicable to and will benefit both frameworks.

## 3. Raw Data Availability: Why Does It Matter?

Performance landscapes are often drawn using extrapolation methods such as thin-plate splines (TPSs) [14]. TPSs uses information from the collected data in each nutrient ratio and concentration to extrapolate the values for the areas of the nutritional space where experimental data is lacking. This allows for the construction of a continuous landscape from discrete experimental data. This approach is widespread to study morphological, behavioural, physiological and reproductive adaptations in species, spanning from insects to mammals [6,7], including humans [15]. However, while the extrapolation method solves the data collection conundrum, it generates a new problem—TPSs often mask the information from the raw (empirical) data, making it virtually impossible to infer and/or extract these data (see, e.g., Figure 1B). As a result, each GFN study becomes an ‘island’ of information inaccessible to researchers. This limitation strongly limits the possibility to perform meta-analyses, and discover broad-scale patterns through comparative studies, which is essentially developing the GFN from a descriptive to an analytical method.

## 4. More Data for A Bigger Picture

GFN studies are increasingly published with nutritional response landscapes in a variety of organisms (see summary in Appendix A) [2]. The data for each population and species are time consuming (and often costly) to obtain but extremely insightful for the community. For example, Lee et al. [7] ran experiments with 1008 fruit flies (28 diets, 36 replicates) to build three landscapes showing for the first time that nutrient balance, and not caloric restriction, was responsible for extending lifespan in insects (Figure 1B). Similar conclusions were later obtained in mammals by Solon-Biet et al. [9], who ran experiments with 858 mice (25 diets). This fast-growing amount of empirical data thus offers the new fascinating possibility to compare traits between populations and species. To date, however, performance surface data have not been provided in more than 60% of the GFN studies (see meta-analysis in Appendix A). We believe that making such raw GFN data open access and published alongside studies (irrespective of the journal’s policy) will allow for the development of a truly integrated and collaborative community of nutritional ecologists with accessible datasets from which quantitative analyses can be performed and new methods for data analysis can be designed and tested [16,17].

## 5. How to Share GFN Data?

Although the value of open data has been widely recognized to explore broader patterns and processes across species, space and time, few fields of biological and ecological sciences have fully embraced this practice, and so the landscape of open data remains largely scattered and complex to navigate [18]. To facilitate access to and reuse of GFN data by the community, we encourage authors to adopt a simple common guideline.

Provide raw data used to construct landscapes in a standardized and accessible format (see example in Figure 1E; see template file (.csv) in Appendix B).If publishing in an open-access journal, authors are likely (although not always) to be required to deposit the raw data in a public data repository (e.g., Dryad). If publishing in a subscription journal, we encourage authors to make raw data available as Appendix A. In both cases, it is important that the data format is also standardised (Figure 1C) and any additional information needed for the understanding of the dataset is also provided (e.g., README file). These steps will benefit the visibility and citations of the original paper.Include ‘geometric framework for nutrition’ and ‘performance landscapes’ as keywords in publications to increase exposure on search engines.If the study is already published but the data are not available, we encourage authors to deposit the raw data in a public data repository.

## 6. Towards Quantitative and Comparative Nutrition Research

Open data have the potential to transform biological and ecological sciences with a new depth of information that can facilitate advances across disciplines and explore broader-scale patterns [19]. Nutritional ecology is not an exception and will assuredly greatly benefit from this practice, since good quality data are particularly hard to obtain. Just like the adoption of performance landscapes has brought new fundamental insights into the ecology and evolution of nutrition [4,6,7,17], formally comparing large quantities of landscapes based on data sharing will allow for new kinds of quantitative analyses characterizing the responses of individuals to nutritional conditions from different life-stages, populations and species. Recent studies have already demonstrated the potential for new theoretical and comparatives avenues of research with open data in nutritional ecology [16,17]. For instance, we developed a quantitative approach to compare GFN performance landscapes within and between species, and revealed major differences in the reproductive responses of *Drosophila melanogaster* (data from [7]) and the tephritid *Bactrocera tryoni* (data from [20]), whereby the species-specific reproductive responses are driven by differences in protein (but not carbohydrate) intake [17]. Likewise, comparative studies on the relative composition of diets using the RMT have provided key insights into the nutritional ecology of higher vertebrates in the wild including marine predators [21] as well as primates (including humans [22,23]; see above). Beyond nutrition research, strictly speaking, these data could be used to study a broad range of biological interactions mediated by food and quantitatively address general problems in ecology and evolution. Are there nutritional adaptations required for sociality? How do nutritional interactions mediate host–parasite evolution? To what extinct do nutritional adaptations shape species assemblages? How do nutritional constraints associated with trophic levels determine evolutionary trade-offs? Many of these questions could be addressed through meta-analyses of open nutritional data.

## 7. Conclusions

Nutritional ecology studies increasingly use nutritional landscapes to map the complex interacting effects of nutrient intake on animal performances, in a wide range of species and ecological contexts. Here we encourage authors to adopt a common guideline to share their raw data. Opening access to these hard-to-obtain, yet considerably insightful data is fundamental to develop a comparative framework for nutrition research and offer new quantitative means to address open questions about the ecology and evolution of nutritional processes.

## Figures and Tables

**Figure 1 insects-11-00236-f001:**
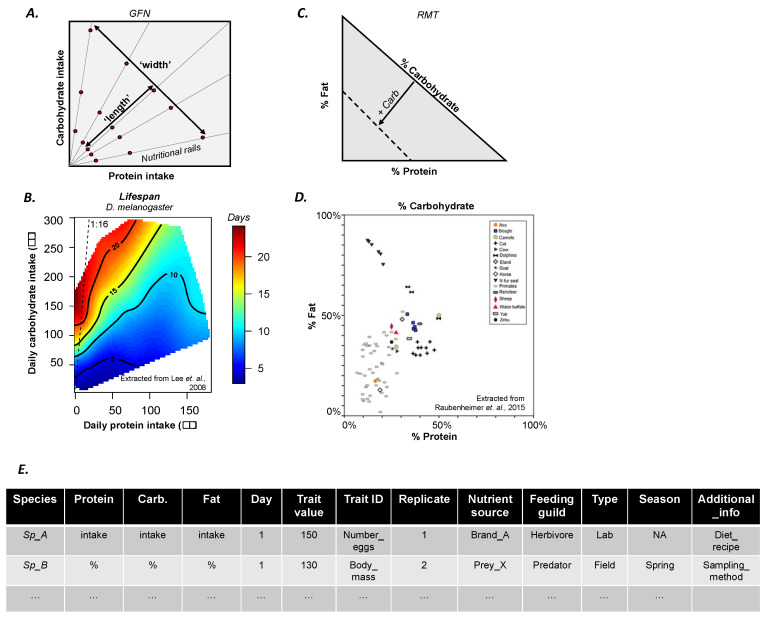
Overview of the geometric framework for nutrition (GFN). (**A**) Schematic representation of the performance landscape plotted in a two-dimensional nutrient space (here defined by protein (P) and carbohydrate (C) intake). Nutritional rails are diets with varying PC ratios. Red dots represent the concentrations of the different diets. The ‘width’ of the landscape is determined by the nutritional rails. The ‘length’ of the landscape is determined both by the concentrations within each rail and the individual variation in nutrient intake within each rail. (**B**) Empirical performance landscape (i.e., lifespan) from a GFN study in *Drosophila melanogaster* (extracted from [7]). (**C**) Schematic representation of the Right-Angle Mixture Triangle (RMT) used to assess diet mixtures primarily in field studies. Note that the % of carbohydrate in the mixture increases as the axis moves towards the origin (solid arrow) so that the mixture is constrained within the limits delimited by the dashed isocline. (**D**) Empirical Right-Angle Mixture Triangle (i.e., milk composition) from a field study (extracted from [12]). (**E**) Template for storing and sharing GFN data. An Excel (.csv) version is provided in Appendix B.

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
