# Peer review of "Open Data for Open Questions in Comparative Nutrition"

_insects, 2020, doi:10.3390/insects11040236_

Round 1

Reviewer 1 Report

This Opinion highlights the importance of making accessible the data generated in studies using the approach of Geometric Framework for Nutrition. It also propose a model to standardised the data and the general steps to follow. I agree with the authors that there is an incredible amount of data that could be used if it was accessible and I do enjoy their initiative. I have few minor comments related to the manuscript.  

Line 14-15: I would rather say ”use nutritional landscapes to map performance”.

Line 39-40: I would say “intakes on performance and ultimately fitness expression” as researchers almost always measure traits individually and then reconstruct the whole picture to understand impact on fitness.

Line 75-77: I think I understand what authors mean when they use degrees and talk about angles but I found this sentence not really clear. In particular non-specialist readers can be really confused. Please clarify it.

Line 85: typo “in” should be replaced by “is”.

Appendix and data table: I think it is also important to include the sources of proteins and carbohydrates (and potentially other nutrients) that are used during experiments (e.g. source of carbohydrates: glucose, sucrose or starch?; protein source: what amino acids?). For a given P:C different molecules can be differentially metabolised and can thus induces different effect on the response variable. Normally this information should already be present in the method section of papers but I believe it is an important parameters to store with the rest of the data.

A last point, I would have like to see an exemple of what kind of study/comparison can be achieve with shared data. The authors mention many time cross species/population comparison and I think an illustrated example could be more appealing for readers.

Reviewer 2 Report

In this manuscript, Morimoto and Lihoreau write an interesting opinion piece on the current state of data accessibility for nutritional studies. They argue for a number of simple guidelines to facilitate access to and allow reuse of nutritional data by other researchers. I found myself nodding in agreement when reading through this manuscript, but I fear that people are often scared of sharing data. That being said, I do think science is changing to being more open and the guidelines put forth in this manuscript would make it easier to perform things like meta-analyses. As such, I only have a few comments that are mainly based on the figures. I am guessing as this is an opinion piece that the number of references is limited, but I would cite Sterner and Elser’s 2002 Ecological Stoichiometry book (“Ecological stoichiometry: the biology of elements from molecules to the biosphere”) on line 35.

Figure 1c: I think the one of the main points of this paper is trying to create a simplified template that people can use to submit their data to digital repositories in a standardized format. However, I think the template provided here could use a bit of work. First, artificial diets are often made up of numerous ingredients and I am not sure a single “Nutrient Source” column is enough to include all the relevant information. I am sure this could be addressed in meta-data or readme files, but I am a bit worried that this would also be covering up a lot of information. Second, the column “Trait value” does not have an additional identifier to what that trait value actually is. I think this would be necessary to actually say what is being measured. Third, often in nutritional studies, multiple traits are measured from a single individual (e.g. fat content, development, mass, etc.). Would the goal be to have a new row for each trait measurement to fit the template -or- just additional columns with the different rows representing individuals. I think the answer to this would affect my second point above. Fourth, this is quite minor but since you have field and lab studies as an option, it might be good to also have the time of year the study was done. That would add some extra information for field biologists as nutritional demands and resource collection can vary throughout the year.

Figure 1b: Is this data from a particular study, collected by the authors, or is it made up? It doesn’t really matter, but if it was pulled from a particular study then I think proper attribution in the figure legend is needed.

Line 77: I think the equation here is unneeded and it should be easy enough to see a 20% increase from the numbers provided.  

Reviewer 3 Report

This is an oponion paper.

The authors argue that while recent statistical techniques for analyzing diet-performance data like thin-plate-splines (Tps) are good at visualizing complex relationships, they also effectively hide the actual measurements (raw data) on which the analyses are based. The paper is a plea for researchers that publish such studies to recognize the value of making raw data available either in data depositories or as e-supplements to the paper. It is difficult not to sympathize with this; in fact, many journals to day require or at least strongly encourage authors to make data available either of these ways. The need for stressing this point comes from the authors’ enquiry of a selection of 69 diet-performance papers of which only 40% had made their data publicly available. If this article can convince more authors to add raw data as e-supplements to articles in journals that do not require this, then it will have served its purpose.

The template for storing and sharing of GFN data (fig. 1c) should  be widened to account for studies on species that are not strictly herbivorous. The reference overview in the supplementary data seems to have overlooked existing studies on predators (where protein and lipid are the most relevant macronutrients), but includes several omnivorous species (e.g. crickets) for which lipids may be as relevant as carbohydrate or even more. Most existing studies treat such species as if they are herbivores (i.e. consider only protein and carbohydrate), but this may not give the full picture of these species’ nutritional ecology. Probably, future studies will increasingly include all three macronutrients. Thus, the diet-ratio column could be P:C:L (protein:carbohydrate:lipid); if only two macronutrients are included in a study, it could be indicated as P:C:- (herbivores) or P:-:L (predators).  

Consequently, the “nutshell”-introduction to GFN (fig. 1ab) should include also the right-angled mixture triangle and an example of a performance study that included all three macronutrients.
